# Transfer learning for predicting acute myocardial infarction using electrocardiograms

**Axel Nyström**[1,2,*], **Anders Björkelund**[2], **Mattias Ohlsson**[2,3], **Jonas Björk**[1,4], **Ulf Ekelund**[5,6], **Jakob Lundager Forberg**[6,7]

1 Department of Laboratory Medicine, Lund University, Lund, Sweden, 2 Computational Science for Health and Environment (COSHE), Centre for Environmental and Climate Science, Lund University, Lund, Sweden, 3 Center for Applied Intelligent Systems Research (CAISR), Halmstad University, Halmstad, Sweden, 4 Clinical Studies Sweden, Forum South, Skåne University Hospital, Lund, Sweden, 5 Department of Emergency and Internal Medicine, Skåne University Hospital, Lund, Sweden, 6 Department of Clinical Sciences, Lund University, Lund, Sweden, 7 Department of Emergency Medicine, Helsingborg Hospital, Helsingborg, Sweden

* axel.nystrom@med.lu.se

**Data availability statement:** This study is based on sensitive individual-level,

## Abstract

At the emergency department, it is important to quickly and accurately identify patients at risk of acute myocardial infarction (AMI). One of the main tools for detecting AMI is the electrocardiogram (ECG), which can be difficult to interpret manually. There is a long history of applying machine learning algorithms to ECGs, but such algorithms are quite data hungry, and correctly labeled high-quality ECGs are difficult to obtain. Transfer learning has been a successful strategy for mitigating data requirements in other applications, but the benefits for predicting AMI are understudied. Here we show that a straightforward application of transfer learning leads to large improvements also in this domain. We pre-train models to classify sex and age using a collection of 840 k ECGs from non-chest-pain patients, and fine-tune the resulting models to predict AMI using 44 k ECGs from chest-pain patients. The results are compared with models trained without transfer learning. We find a considerable improvement from transfer learning, consistent across multiple state-of-the-art ResNet architectures and data sizes, with the best performing model improving from 0.79 AUC to 0.85 AUC. This suggests that even a simple form of transfer learning from a moderately sized dataset of non-chest-pain ECGs can lead to major improvements in predicting AMI.

## Author summary

In this paper, we focus on machine learning models that use electrocardiograms (ECGs) to detect heart attacks among patients presenting with chest pain at the emergency department. The performance of such models depends strongly on the availability of

pseudonymized, healthcare data from the Skåne Emergency Medicine (SEM) database (https://doi.org/10.1186/s13049-024-01206-0). The data thus contain potentially identifying or sensitive patient information and cannot be made publicly available for ethical and legal reasons (c.f. the Public Access to Information and Secrecy Act, as well as the General Data Protection Regulation). However, we welcome initiatives on international collaborative projects. Anonymized parts of the database can be made available for researchers upon reasonable request, although this may require additional ethical permits. Inquiries can be sent to Ulf Ekelund (ulf.ekelund@med.lu.se) at Lund University or to Clinical Studies Sweden, Forum South (halsodata.sus@skane.se). The code used in this study is available at https://www.github.com/Tipulidae/mim.

**Funding:** This study received funding from the Swedish Research Council (https://www.vr.se/english.html, VR; grant no. 2019-00198, awarded to JB), the Swedish Heart-Lung Foundation (https://www.hjart-lungfonden.se/, grant no. 2018-0173, awarded to UE), and Sweden's innovation agency Vinnova (https://www.vinnova.se/en/, grant no. 2018-0192, grant awarded to JB). The funders had no role in study design, data collection and analysis, decision to publish, or preparation of the manuscript.

**Competing interests:** The authors have declared that no competing interests exist.

high-quality examples where the outcome is known. However, gaining access to enough data is difficult and expensive. In this paper, we demonstrate the effectiveness of a type of workaround known as transfer learning. The idea is based on the observation that ECGs of patients who do not suffer from chest pain are relatively common and easy to obtain. We show that using these other ECGs to predict age and sex as a sort of "warm-up exercise" will make the subsequent task of predicting heart attacks considerably easier. In other words, the very process of guessing someone's age forces the algorithm to identify patterns in the ECG that are also useful for predicting heart attacks. In the paper we quantify the performance increase, which is substantial, and how it depends on the size of the datasets. We believe that techniques such as these have the potential to dramatically improve the performance of ECG-based decision support tools in the emergency care setting.

## Introduction

Acute myocardial infarction (AMI) is the main concern for patients with chest pain at the emergency department (ED). Early detection of AMI is crucial for initiating timely treatment and to reduce mortality and morbidity [1]. Yet fewer than 10% of chest-pain patients are diagnosed with AMI, which presents a challenging problem for ED physicians [2]. In order to accurately diagnose or rule out AMI, patient management includes the interpretation of electrocardiograms (ECGs), blood tests, and often prolonged and costly observations.

The ECG is one of the cornerstones of diagnosis of AMI, being fast, cheap, non-invasive and widely available across different medical facilities. The interpretation of an ECG requires comprehensive training however, with notable variability in diagnostic skills among physicians, across different medical facilities, and outside regular office hours [3]. This inconsistency extends to expert cardiologists, highlighting a broad variance in detecting various heart conditions [4]. To assist clinicians, automated ECG analysis has been evolving for many years, accelerated by advances in machine learning (ML) and the digitization of ECGs and other healthcare records [5,6].

Machine learning (ML) in general and deep learning in particular almost always profit from larger training datasets [7]. Large models can learn more complex patterns, but require more data to achieve good results compared to smaller models [8]. Acquiring more data is among the most straightforward and reliable ways of increasing model performance, but doing so can be both difficult and expensive. In the pursuit of better models to predict AMI using ECGs, the standard approach is to train models using ECGs where the outcome (AMI) is known. But ECGs are also routinely collected outside the ED and for reasons other than trying to detect short term AMI.

Transfer learning is a collection of techniques through which a *downstream*, also known as *target*, task is improved by means of first learning an *upstream*, or *source*, task in a step called pre-training. The pre-trained model is then fine-tuned on the target task, with the expectation that some of what was learned from the source task will generalize—transfer—to the target task, thereby achieving higher performance on the target task. There are multiple variations of transfer learning, but typically the source task utilizes a different dataset and/or a different outcome.

In this paper, we propose to use the predictions of age and sex as pre-training tasks, either individually or in combination, for the downstream task of predicting AMI within 30 days. Unlike potentially stronger, more clinically relevant labels that might be used for pre-training, age and sex are virtually always available, and previous research have shown that both labels

can be accurately predicted from ECGs [9–11]. Using a target and source dataset of 44 k and 840 k ECGs respectively, we show that pre-training on age and sex offers a substantial improvement in terms of AUC for predicting AMI. This improvement is consistent across several state-of-the-art models described in the recent literature.

We further analyze the role of the sizes of source and target datasets by retraining the models with progressively smaller training datasets. We show both that (a) in a small-target data setting where the size of the target training set is reduced to 10% (2.5 k ECGs), the gap to training with the full target dataset can be entirely bridged by pre-training. Conversely (b), assuming the full target dataset for training (25 k ECGs), the AUC on AMI prediction can be improved from 0.79 to 0.85 through transfer learning using 840 k ECGs that are unrelated to AMI but are coupled with age and sex information, which must be regarded as a considerable improvement.

## Related work

Transfer learning has become a staple in several fields where ML is applied, particularly image analysis. Only recently has transfer learning also been successfully applied to ECG classification. A possible reason for the relatively slow adaptation may be the lack of sufficiently large open ECG datasets, which remains an obstacle in the field even now.

Efforts to utilize transfer learning for ECG classification can be broadly divided into two categories. The first uses the raw ECG signals and the second uses image-based methods in which models are pre-trained on large image datasets (usually nonmedical images) such as ImageNet and fine-tuned on ECGs converted to images (typically either as spectrograms or time-series plots) [12,13]. Here we limit our overview to the former approach.

In 2020, Strodthoff et al. [10] published one of the first successful results of transfer learning on ECGs. The authors used data from two public ECG databases (PTB-XL [14], containing 21 837 12-lead ECGs, and ICBEB2018 [15], containing 6 877 12-lead ECGs) and found significant improvements in terms of macro AUC for a collection of 71 diverse outcomes.

Also in 2020, Jang et al. [16] used 2.6 M unlabeled single-lead ECGs to pre-train an autoencoder, which was then fine-tuned on 10 k ECGs to predict a collection of 11 different ECG rhythms.

In 2021, Weimann et al. [17] showed consistently improved performance on atrial fibrillation (AF) detection on the PhysioNet/CinC 2017 dataset [18] through the use of transfer learning. Their pre-training on the Icentia11k dataset [19] consisted of over 630 k hours of single lead ECG data with labeled heart beats. Their ResNet models also showed good performance when evaluated on the PTB-XL dataset.

In 2022, Mehari et al. [20] reported success with a self-supervised-learning algorithm called Contrastive Predictive Coding (CPC). This can be viewed as a form of inductive transfer learning where the source and target datasets are the same, but the source labels are automatically induced from the data itself without human annotation [21]. The authors found that pre-training improved the downstream performance on PTB-XL in terms of macro AUC, data efficiency and robustness against input perturbations.

Following the thread of self-supervised learning, in 2023, Mehari et al. [22] combined a type of model called structured state space models (S4) with the CPC architecture for self-supervised learning, improving state-of-the-art on the PTB-XL dataset. The authors also indicated that the addition of patient metadata (specifically age, sex, height, and weight) alongside the ECG signal further improved downstream performance on PTB-XL.

These results are all promising indications that transfer learning can lead to improvements for a large range of classification tasks and pre-training scenarios. However, most related work either does not focus on AMI, or only includes AMI together with many other outcomes, as is the case in the PTB-XL outcomes.

## Materials and methods

### Data sources

Our data source was the Skåne Emergency Medicine (SEM) cohort, which contains consecutive ED visits between 2017-01-01 and 2018-12-31 from eight different hospitals in Skåne, Sweden [23]. The SEM cohort includes all ED visits from all adult (over 18 years) patients with a Swedish personal ID number, excluding only patients leaving against medical advice, or who have requested to be removed from the dataset. In this study, we further excluded patients without ECGs of sufficient technical quality, as well as patients without any diagnoses registered in the Swedish national patient register.

Besides data from the ED visits, each patient has a five-year medical history of diagnoses and ECGs from all health care visits, collated from comprehensive national and regional registers. The ECGs are all 10 s, 12-leads, sampled at 500 Hz with 16-bit precision.

Of the twelve leads, we only use leads V1-V6, I, and II (in that order), since the remaining four can be constructed as linear combinations of the others. The models were adapted where necessary to comply with this standard, to simplify comparisons.

In the following sections, we describe how the source and target datasets were constructed from the SEM cohort and divided into training, validation and test sets. The overall process and exclusion criteria are shown in Fig 1.

**Target data.** The target dataset consists of ECGs from patients arriving at the ED with chest pain as primary complaint, and the label is the binary outcome of AMI within 30 days of arrival at the ED, defined as the AMI diagnosis (ICD-10 code I21) being set in the Swedish national patient register [24]. We focus on the area under the curve of the receiver operating characteristic (AUC) as our evaluation metric for AMI predictions. We use a single ECG per patient visit, chosen as the first ECG within 3 h of arrival, or if there is no such ECG, the last ECG within 2 h prior to arrival. The reason for reverting to ECGs from before ED arrival is to catch ECGs recorded by paramedics or at the ED before registration.

The target dataset contains 44 370 visits from 37 447 patients, partitioned into training, validation and test sets. The partitioning was done in a *patient-wise* manner, such that records from the same patient were always placed in the same split. The test set was further divided into a random and a temporal split, allowing us to evaluate the performance both on data with the same underlying distribution as the training data (the random test split) and on data which is separated in time from the training data (the temporal split). The temporal test split consists of all visits after a cut-off date of 2018-10-05, which was chosen such that the temporal split would contain 15% of the patients in the whole target dataset. Visits from patients in the temporal test set prior to the cut-off date were excluded (2 717 visits from 1 276 patients). The training, validation and random test sets were constructed as a patient-wise random split from patients that were not included in the temporal test split, with 55% of the patients being in the training set, 15% in the validation set, 15% in the random test set, and the final 15% of the patients being in the temporal test set.

**Source data.** The source dataset contains 836 972 ECGs from 162 903 patients in the SEM cohort, which includes all ECGs from the five-year patient history, not just those associated with ED visits. We excluded all ECGs from all patients in the target dataset in order to minimize the interaction between the two datasets. There is, in other words, no patient overlap

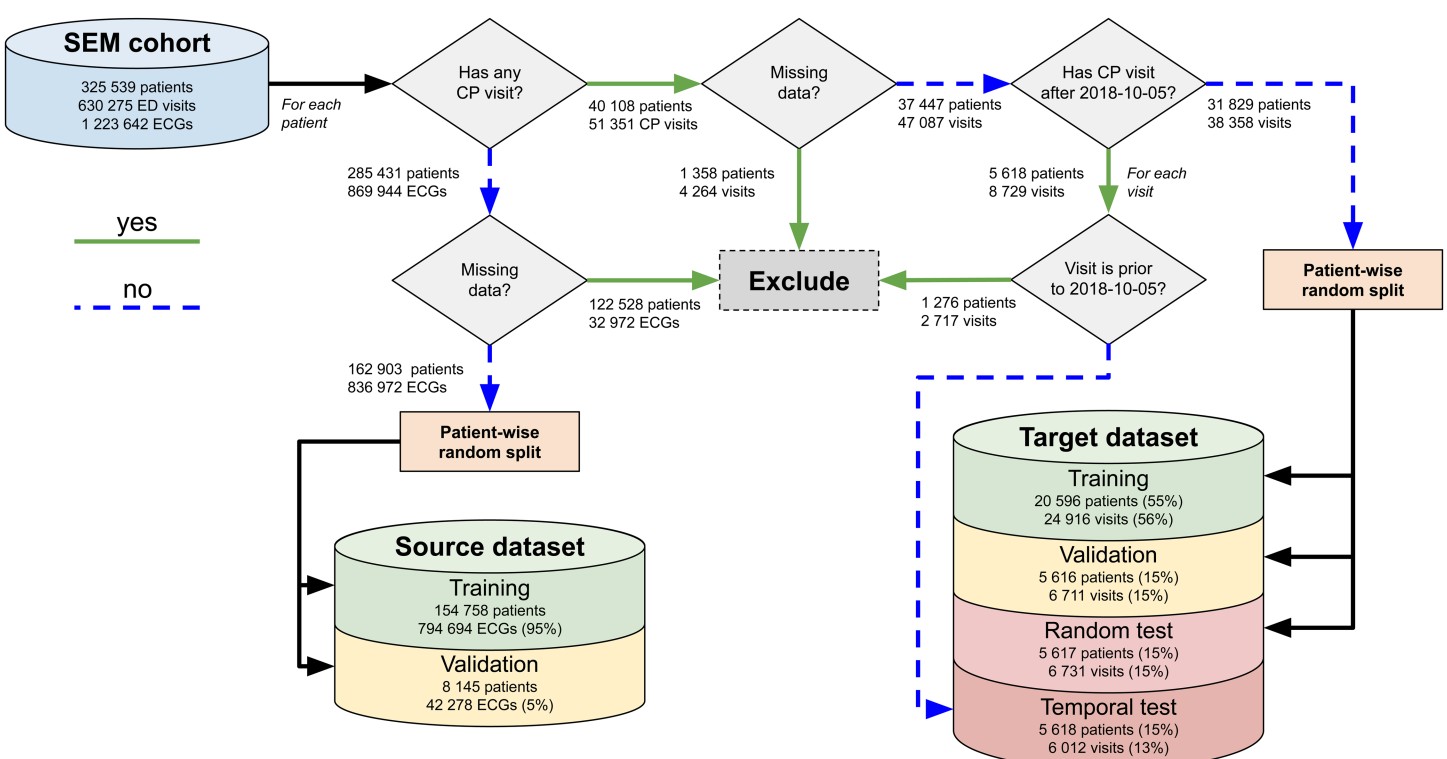

**Fig 1. Visualization of exclusion criteria.** SEM refers to the Skåne Emergency Medicine cohort, which includes all ED visits between 2017-01-01 and 2018-12-31 from 8 hospitals in Skåne, Sweden [23]. The target dataset is based on chest-pain visits, and divided into training, validation and test sets in a patient-wise manner. The test set is divided into a random and a temporal set, with the temporal set including all visits after a cutoff date (2018-10-05). Visits prior to the cutoff were excluded for patients in the temporal test set. The source dataset includes all ECGs from patients that were not included in the target dataset. CP = chest pain, ED = emergency department.

between the source and target datasets. An additional 122 528 patients were excluded due to missing or corrupted data, the majority (114 387) of which did not have any ECGs on record. The source dataset was partitioned into a training and validation set using a patient-wise random split, with 5% of the patients in the validation set and the rest in the training set. We did not hold out a test set for the source data, since our primary concern is the final performance on the downstream task, rather than the generalizability of the pre-training task itself. For the supervised pre-training task, we used age and sex as labels. We used the mean absolute error (MAE) as primary evaluation metric for age regression, and AUC as primary evaluation metric for sex classification.

The age and sex distributions for each dataset and split are summarized in Table 1, which also shows the incidence of AMI for the target dataset splits. The sex distribution is close to 50% for the source data, but the target dataset has more visits from men. Furthermore, the patients in the source dataset are roughly 10 years older than those in the target dataset. These observations are not particularly surprising: men are generally more exposed to cardiovascular disease than women, and we expect general health-care visits to be more strongly correlated with age than chest-pain ED visits. Finally, we note that the incidence of AMI is fairly low, at under 6%.

**Ethics statement.** The creation of the SEM cohort and its use for ML research has been approved by the Swedish Ethical Review Authority (Dnr 2019–05783) and Region Skåne (KVB 302-19). All included patients had access to written information on the SEM cohort and its purpose, and had the possibility to decline participation at any time, for any reason.

**Table 1. Patient characteristics.**

| Patient characteristics | Source dataset | | Target dataset | | | |
|---|---|---|---|---|---|---|
| | Train | Validation | Train | Validation | Random test | Temporal test |
| Patients | 154 758 | 8 145 | 20 596 | 5 616 | 5 617 | 5 618 |
| Visits | N/A | N/A | 24 916 | 6 711 | 6 731 | 6 012 |
| ECGs | 794 694 | 42 278 | 24 916 | 6 711 | 6 731 | 6 012 |
| Female sex, % | 50.4 | 49.7 | 48.0 | 47.1 | 47.8 | 48.3 |
| Age, median (IQR) | 71 (57–81) | 71 (57–81) | 61 (46–75) | 61 (46–74) | 61 (46–75) | 61 (45–74) |
| AMI, % | N/A | N/A | 5.76 | 5.72 | 5.81 | 5.81 |

ECG = Electrocardiogram, IQR = Inter-quartile range, AMI = acute myocardial infarction.

## Models

In this work we consider four different convolutional model architectures applied to 10 s, 12-lead ECGs described in the literature: a small model from our own prior research that serves as a baseline model [25], together with three variations of residual neural networks (ResNets) of different size and complexity. The selected models (except our baseline model) achieve state-of-the-art performance on their respective tasks. Apart from an initial calibration of learning rate, batch size and number of epochs, we did not perform any parameter tuning for the models under consideration. Unless otherwise noted, the code implementation of the models were adapted from the respective author's public github repositories (with some cleanup and refactoring).

A brief overview of the models and their complexity in terms of number of parameters and convolutional layers is shown in Table 2. A more detailed description of the model parameters, architectures, and training times can be found in Sects C, D, and E in S1 Appendix. In the following sections, we describe the origins and general structure of each model.

**CNN-20k.** This was the best performing model from [25] using only the raw input ECG to predict major adverse cardiac events (MACE) within 30 days of arrival at the ED. It contains two convolutional layers followed by two fully connected layers. Although small compared to the other models, it should be noted that this model was tuned for a very similar task (the majority of MACE outcomes are due to AMI) on a very similar dataset (ESC-TROP, which is a subset of SEM).

**RN-900k.** The xresnet1d50 (RN-900k) model [20] is an adaptation of the original ResNet-50 model from [28], which also incorporates a number of tricks from [29]. The model consists of 51 convolutional layers arranged into four stages, where each stage further contains 3–6 residual blocks. The residual blocks each contain 3 convolutional layers and a shortcut connection. The RN-900k model was found to outperform other ResNet variations on the PTB-XL dataset for a variety of tasks. In particular, the model has been fine-tuned to perform well on the macro AUC score of 71 different outcomes from PTB-XL, including AMI.

**Table 2**. Model summary.

| Model | Conv. layers | Parameters | Source |
|---|---|---|---|
| CNN-20k | 2 | 20 479 | Nyström et al. [25] |
| RN-900k | 51 | 892 449 | Mehari et al. [20] |
| RN-7M | 9 | 6 784 561 | Ribeiro et al. [26] |
| RN-33M | 25 | 33 062 569 | Gustafsson et al. [27] |

Overview of included models and their complexity in terms of parameters and convolutional layers.

The model also incorporates a type of data augmentation and ensembling: During training, the 10 s input ECG is randomly cropped to a 2.5 s long sequence (random sliding windows), whereas during validation and testing, the input ECG is sliced into 10 equidistant, overlapping 2.5 s slices, and the predictions for all 10 slices are aggregated to a single prediction. We consider this augmentation/ensembling scheme as part of the model itself. The other models do not use augmentation or ensembling.

**RN-7M.** The RN-7M model [26] is a ResNet style model based on the architecture described by [30] and was developed and tuned using over 2 M ECGs to recognize six types of ECG abnormalities. The model consists of 9 convolutional layers arranged into 4 residual blocks. Compared to the RN-900k model, this model is "wide and shallow": each convolution uses more filters, has a larger kernel size and downsamples the input signal less aggressively, but there are far fewer layers.

The model downsamples the input ECG from 500 Hz to 400 Hz, and adds zero-padding to make each ECG 4 096 samples along the time-axis. We adjusted it slightly to use only 8 leads instead of the full 12-lead input that was described in the article.

**RN-33M.** The RN-33M model [27] is an extension of the RN-7M model, consisting of 25 convolutional layers arranged into 12 residual blocks. It was developed using 500 k ECGs from the ED to distinguish between STEMI, NSTEMI and non-AMI. The main difference from the RN-7M model besides the number of layers is the addition of a Squeeze-and-Excitation block within each residual block, which is intended to help weighting the channel-wise information [31]. Although still only using half the number of convolutions as the RN-900k model, the RN-33M model is by far the largest in terms of number of parameters at just over 33 M.

Similarly to the RN-7M model, the input ECGs are downsampled to 400 Hz and employs zero-padding to make the signal 4 096 samples wide.

The code for the RN-33M model was not publicly available at the time of writing, but because it is described as an updated version of the RN-7M model, for which code was available, we based our implementation on the RN-7M code and followed the described architectural changes as closely as possible.

## Transfer learning strategy

In order to simplify the analysis, we used the same general transfer learning strategy for each of the four models under consideration, and repeated the process of pre-training and fine-tuning on varying proportions of the available source and target data.

The transfer learning strategy, illustrated in Fig 2, consisted of three stages: In the first stage, a model was pre-trained on the source dataset to predict age, sex, or both, saving the model weights after each epoch. In the second stage, we loaded the weights from the epoch where the best validation loss was observed. The classification head (i.e. the final layers of the model, everything after the flatten layer) was then replaced by a fully connected layer of size 100, followed by a dropout layer with $p = 0.3$, followed by a fully connected output layer with sigmoid activation. The replacement classification head was randomly initialized, and training proceeded on the target dataset to predict AMI. During the second stage, the weights of the model body were frozen, so that only the weights of the classification head were updated during backpropagation. Finally, in the third stage, fine-tuning continued with all model weights unfrozen.

For the first two stages we used a one-cycle learning rate schedule consisting of an initial linear warmup, followed by a cosine decay with a factor of 100, so the learning rate at the end

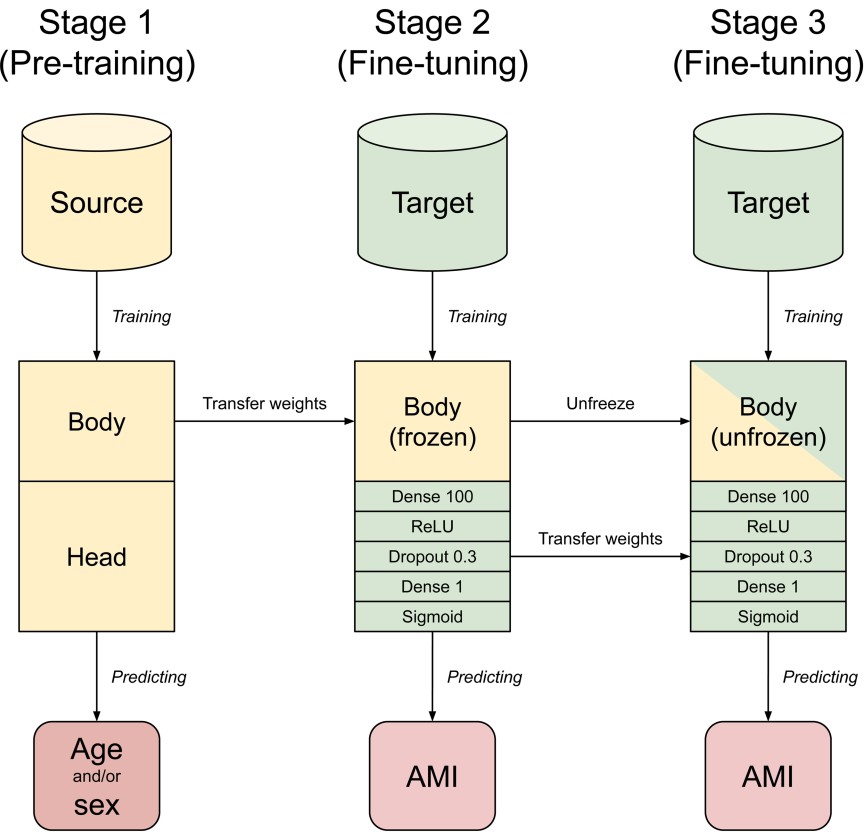

**Fig 2. Visualization of the transfer learning strategy.** The model is divided into two segments, which we call the body and the head. In the second stage, the head of the pre-trained model is replaced by a small neural network with a single hidden layer. The model body is frozen in stage 2, meaning that the parameters are not updated during back-propagation.

is two orders of magnitude smaller than the warmup target. The final stage used a constant learning rate.

For the target task, we used binary cross-entropy (CE) as the loss function, and performed early stopping where the AUC on the validation set was maximized. For the source task loss function, we used CE for sex prediction and MAE for age prediction. When predicting both age and sex simultaneously, we used a weighted sum of the MAE of age and CE of sex as the loss, where the weight was chosen as 1 for CE and 0.045 for MAE. These weights were chosen empirically from early experiments such that each target would contribute roughly equally to the total loss.

The model parameters and architectures are described in more detail in Sects C and D in S1 Appendix.

## External validation on PTBXL

To assess the generalizability of the results, we used the PTBXL ECG database [14] as a form of external validation. The PTBXL database contains 21 748 ECGs with detailed annotations for a range of pathologies, including AMI. The data is split into ten pre-defined folds, where the ninth and tenth folds have a higher degree of certainty in the labels, and are meant to be used for validation and testing.

We evaluated our transfer learning strategy on the PTBXL dataset in two ways: pre-training on SEM and fine-tuning on PTBXL, and pre-training on PTBXL and fine-tuning on SEM. In the former case, we fined-tuned using the aggregate AMI diagnosis (diagnostic class MI) as outcome, for which the incidence was 25.1%. When using PTBXL for pre-training, we used patient age as the target outcome, and the median age was 62 years (IQR 50 – 72).

## Shortcut learning

A recent concern about deep learning models is the risk of shortcut learning, which is when a model learns unintended patterns that generalize poorly [32]. To assess the presence of short-cuts from potential differences between ECG machines, we gathered a number of metadata variables, including brand, software version, location, and cart ID, and used them to attempt to predict the outcome.

Another potential shortcut is in the pre-training task itself, i.e. age and sex. Although both age and sex are useful predictors for AMI on their own, their usefulness for transfer learning would be diminished by the extent to which we could get similar improvements by simply adding age and sex as input features directly. In other words, it could be that the observed improvement from our transfer learning strategy is merely a result of the models being able to infer useful features (age and sex), rather than a more general ability to interpret ECGs. This would constitute an unwanted case of shortcut learning, because the gains from transfer learning would be nullified by the introduction of simple features that were easily available all along. We test this potential shortcut by fitting logistic regression models to age and sex together with the *predictions* from the ECG models, either with or without transfer learning. If logistic regression performs similarly with the pre-trained model predictions as it does with predictions from models trained from scratch, this would indicate that age and sex were shortcuts, greatly diminishing the usefulness of our transfer learning approach.

## Fairness

We investigate model fairness with respect to age and sex by stratifying the results on each subgroup separately. Although this approach has the potential to reveal systematic differences between groups, it is not enough to elucidate the reasons for any observed differences. Both age and sex are known to correlate with the AMI outcome, and this may play a role in the effect of transfer learning on these subgroups.

## Results and discussion

### Main results

The overall AUC on the target task for each model is shown in Table 3, with the two test sets shown separately. There is a consistent improvement in AUC for the pre-trained models compared to the models without pre-training. The baseline CNN-20k model displays the smallest improvement overall, ranging from 1 to 4 percent absolute, the smallest when pre-trained on combined age and sex outcome and the largest when pre-trained for sex. The other models display the greatest improvements (up to 7 percent absolute) when pre-trained on age or the combination of age and sex, with sex appearing to be slightly behind. All the models perform worse (about 1 percent absolute) on the temporal test set compared to the random test set. Aside from the baseline CNN-20k model, the improvement from pre-training is similar between the two test sets. The CNN-20k model performed particularly poorly on the temporal test set without pre-training, which leads to an apparently greater improvement from pre-training when compared to the random test set.

**Table 3. Main results.**

| Model | Test set | No pre-training | Age | Sex | Age & Sex |
|---|---|---|---|---|---|
| CNN-20k | Random | 0.785 (0.764 – 0.807) | 0.798 (0.777 – 0.820) | 0.806 (0.786 – 0.828) | 0.794 (0.772 – 0.816) |
| | Temporal | 0.750 (0.724 – 0.775) | 0.778 (0.755 – 0.802) | 0.792 (0.768 – 0.818) | 0.775 (0.751 – 0.799) |
| | Combined | 0.768 (0.752 – 0.785) | 0.789 (0.774 – 0.804) | **0.800** (0.785 – 0.815) | 0.785 (0.770 – 0.801) |
| RN-900k | Random | 0.799 (0.777 – 0.823) | 0.857 (0.839 – 0.875) | 0.845 (0.827 – 0.866) | 0.858 (0.840 – 0.876) |
| | Temporal | 0.787 (0.760 – 0.814) | 0.846 (0.826 – 0.868) | 0.836 (0.815 – 0.857) | 0.844 (0.823 – 0.865) |
| | Combined | 0.793 (0.776 – 0.810) | **0.852** (0.838 – 0.865) | 0.841 (0.826 – 0.855) | 0.851 (0.838 – 0.866) |
| RN-7M | Random | 0.796 (0.774 – 0.820) | 0.852 (0.834 – 0.870) | 0.832 (0.811 – 0.852) | 0.830 (0.809 – 0.850) |
| | Temporal | 0.771 (0.744 – 0.799) | 0.819 (0.799 – 0.842) | 0.808 (0.785 – 0.833) | 0.817 (0.795 – 0.839) |
| | Combined | 0.784 (0.766 – 0.803) | **0.837** (0.823 – 0.851) | 0.821 (0.804 – 0.836) | 0.824 (0.808 – 0.837) |
| RN-33M | Random | 0.729 (0.705 – 0.756) | 0.797 (0.777 – 0.818) | 0.774 (0.751 – 0.797) | 0.796 (0.774 – 0.818) |
| | Temporal | 0.716 (0.687 – 0.743) | 0.778 (0.754 – 0.805) | 0.770 (0.746 – 0.796) | 0.792 (0.769 – 0.816) |
| | Combined | 0.723 (0.703 – 0.742) | 0.788 (0.771 – 0.804) | 0.772 (0.756 – 0.788) | **0.794** (0.779 – 0.810) |

AUC on the test sets for each model, for each combination of pre-training labels. 95% confidence interval in parentheses is approximated with basic bootstrapping [33], with $B$ = 1000 bootstrap samples. The combined test set is the micro-averaged results of the random and temporal test sets. The best result on the combined test set for each model is highlighted in bold.

Overall, the benefit of transfer learning is clear, and the trend is similar across both test sets. The effect is also consistent across multiple performance metrics; for more information, we refer to Sect A in S1 Appendix. The best choice of pre-training label is somewhat inconclusive, with a slight edge towards age or a combination of age and sex.

The pre-training results on the validation set are summarized in Table 4. All three ResNet models perform roughly similar on both pre-training tasks, irrespective of whether they are trained towards a single target or both. The MAE for predicting age lies in the range 6.6 – 6.8 years, and the accuracy of 88–90% for predicting sex. These numbers are in line with previous studies; for instance, Attia et al. achieved an MAE of 6.9 for age regression and an accuracy of 90.4% for sex classification [9]. Strodthoff et al. obtained an MAE of 6.8 years for age regression for healthy patients and 7.2 years for non-healthy patients in PTB-XL, as well as an accuracy for sex classification of 81% for healthy and 90% for non-healthy patients [10]. Lima et al. achieved an MAE of 8.4 years for predicting age [11].

The baseline CNN-20k model performs reasonably well on the sex-classification task, but not so well on the age regression, and struggles especially on the combined task. In contrast, the ResNet models are able to learn both tasks at once with little degradation.

**Table 4. Pre-training results.**

| Model | Label | MAE | $r^2$ | AUC | Accuracy |
|---|---|---|---|---|---|
| CNN-20k | Age | 10.32 | 0.50 | | |
| | Sex | | | 0.92 | 0.81 |
| | Age & Sex | 15.81 | -0.05 | 0.89 | 0.80 |
| RN-900k | Age | 6.69 | 0.76 | | |
| | Sex | | | 0.96 | 0.88 |
| | Age & Sex | 6.80 | 0.75 | 0.96 | 0.88 |
| RN-7M | Age | 6.64 | 0.75 | | |
| | Sex | | | 0.96 | 0.89 |
| | Age & Sex | 6.81 | 0.75 | 0.96 | 0.90 |
| RN-33M | Age | 6.60 | 0.75 | | |
| | Sex | | | 0.96 | 0.89 |
| | Age & Sex | 6.70 | 0.74 | 0.96 | 0.89 |

MAE = mean absolute error, $r^2$ = coefficient of determination.

To simplify the presentation, the next section will focus on the results using age as the pre-training task, and consider only the results on the full test set (i.e. the combined random and temporal test sets). Specifically, the test sets are concatenated and metrics are computed on the concatenation, thus resulting in micro-averages.

## Dependency on dataset size

In order to determine how the size of the target dataset influences the benefit of transfer learning, we fine-tuned each model on progressively larger subsets of the target dataset. Specifically, we consider subsets of 10%, 20%, 30%, ..., 80%, 90% of the target training data, where each set is a strict subset of the next one, chosen randomly at the patient level, so that all examples from a single patient are included at a once. The validation and test sets remain the same for each subset. The results on the (combined) test set are illustrated in Fig 3.

Unsurprisingly, more target data leads to overall better performance but with a reduced benefit from transfer learning. Without pre-training, the CNN-20k baseline model performs similarly to the much larger ResNet models, except for the largest model (RN-33M) which struggles with overfitting and performs much worse than the others. The RN-33M model is clearly too large for our small target dataset. When we introduce transfer learning, however, the RN-33M model improves enough to catch up with the baseline CNN-20k model, but both are still behind the other two ResNet models. With pre-training, it seems like the baseline is too small, and the RN-33M model is too large. The RN-900k and RN-7M models perform roughly the same with a sizable improvement over the other two models.

In order to determine how the effectiveness of transfer learning depends on the size of the pre-training dataset, we pre-trained each model to predict age on 14 overlapping subsets of

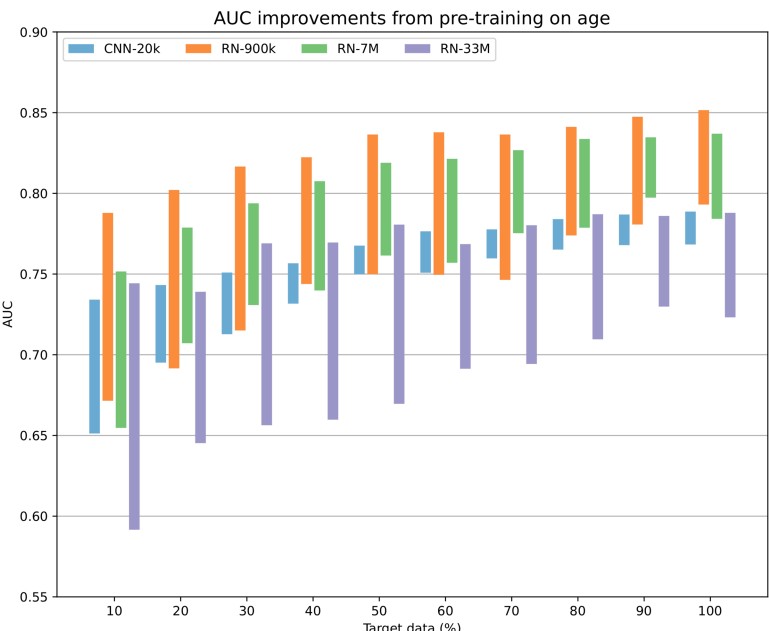

**Fig 3. Bar chart showing the AUC for predicting AMI, micro-averaged over both test sets, for different sizes of the target training set.** The bottom of each bar shows the AUC for models without pre-training, and the top of each bar shows the AUC for models pre-trained on age, using the full source dataset. The size of each bar thus corresponds to the improvement from transfer learning.

sizes 2%, 4%, 6%, 8%, 10%, 20%, 30%, ..., 80%, 90%, 100%, where the full training dataset consists of 795 k ECGs. Each pre-trained model was subsequently fine-tuned to predict AMI for each of the 10 subsets of the target data. The AUC (micro-averaged over both test sets) on the target task (predicting AMI) for the RN-900k model is shown for each combination of dataset size in Fig 4.

Although there are some exceptions due to the stochastic nature of the optimization procedure, the general trend is clear and expected: The more data the better the final results, regardless of whether that data is in the source or target domain or some combination of the two. In particular, a 90% reduction in the size of the target dataset can be compensated for by pre-training with approximately 800 k ECGs. Most of the improvement from pre-training is realized already with relatively small source datasets, although this effect is more pronounced when the target dataset is also small. For instance, the improvement from pre-training on 80 k ECGs is between 9 percentage points (for small target sets) and 3 percentage points (for large target sets), but increasing the pre-training dataset by an order of magnitude only yields an additional improvement of roughly 3 percentage points regardless of the size of the target dataset. An intuitive explanation for this is that in the low-data domain, the pre-training task primarily makes up for rudimentary patterns that are not as impactful when the target dataset is larger.

## Pre-training on SEM, fine-tuning on PTBXL

We fine-tune tuned on the AMI outcome in PTBXL, using models pre-trained on age with the full source dataset from SEM. The pre-training was done in the same way as when fine-tuning on SEM. The results are shown in Table 5.

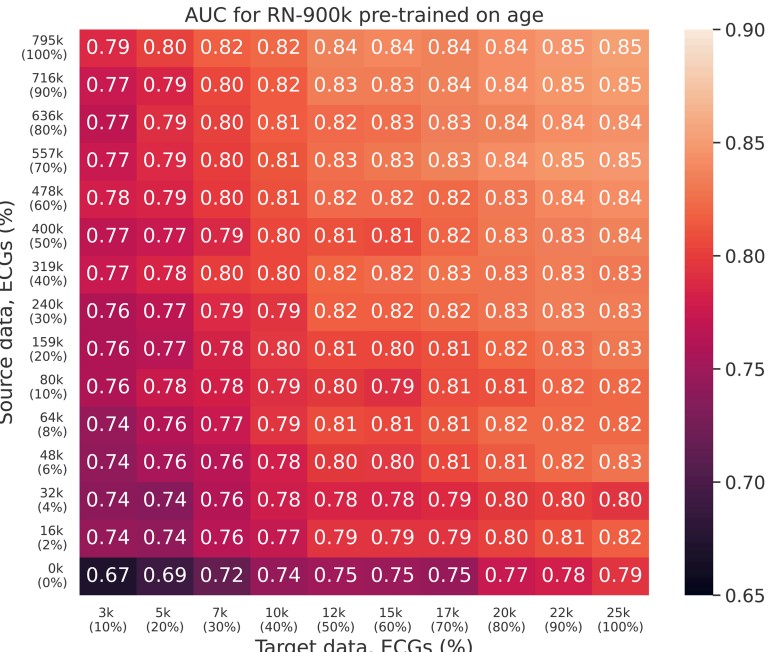

**Fig 4. Heatmap showing the AUC for predicting AMI with the RN-900k model, for different amounts of source and target training data.** The x-axis shows the number of ECGs in the target training dataset, and the y-axis shows the number of ECGs in the source training dataset. The bottom row shows results without pre-training.

**Table 5. Predicting AMI in PTBXL.**

|          | From scratch | Fine-tuned |
|----------|--------------|------------|
| **CNN-20k** | 0.903 | 0.917 |
| **RN-900k** | 0.924 | 0.927 |
| **RN-7M** | 0.918 | 0.929 |
| **RN-33M** | 0.891 | 0.893 |

Results (AUC) for predicting AMI in PTBXL (on fold 10). Fine-tuned column shows results when using model pre-trained to predict age with the full source dataset from SEM. From scratch are results when training on PTBXL (folds 1-9) without any transfer learning.

All the models benefited from the pre-training on SEM, demonstrating that the transfer learning strategy generalizes between datasets. The magnitude of the effect was somewhat reduced in PTBXL compared to SEM, which is to be expected considering the differences between the populations.

In general, all models performed better on the AMI task in PTBXL than they did in predicting 30 day AMI in SEM. One possible explanation relates to the age of PTBXL, where the majority of the ECGs are from the 1990s. The tools for detecting and diagnosing heart damage, including AMI, has seen a lot of improvements in the past few decades, especially in the form of high-sensitivity cardiac troponin lab assays, which enable the detection of much smaller infarctions. As such, we expect the AMIs in PTBXL to be easier to identify than those in the much more recently collected SEM cohort. Additionally, the incidence of AMI is several times higher in PTBXL compared to SEM, which may contribute to making it an easier learning task.

## Pre-training on PTBXL, fine-tuning on SEM

We used PTBXL to pre-train the RN-900k model on age, and fine-tuned it to predict 30 day AMI in SEM. We performed the fine-tuning step for different sizes of the target dataset. The results, summarized in Fig 5, can be compared to models trained from scratch, and models pre-trained on SEM. In particular, we note that the size of the PTBXL database (17 k ECGs) is similar to the 2% slice of SEM (16 k ECGs) previously considered in Fig 4.

Similarly to pre-training on SEM, there is a consistent and meaningful improvement on the downstream task when pre-training on age in PTBXL. The effect is comparable to the improvement seen from pre-training on a similarly sized slice of the SEM source database,

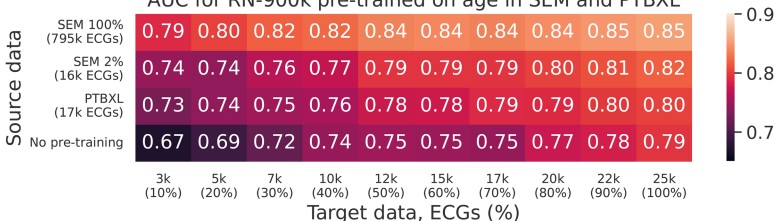

**Fig 5. Heatmap showing the AUC for predicting AMI on SEM with the RN-900k model, for different amounts of source and target training data.** The x-axis shows the number of ECGs in the target training dataset (SEM), and the y-axis shows the number of ECGs in the source training dataset (SEM or PTBXL). The bottom row shows results without pre-training.

suggesting a robustness with respect to the origin of the source dataset that greatly increases the potential utility of the transfer learning strategy.

## Subgroup analysis

We performed a subgroup analysis in which the test set was stratified on age quartiles and sex. In other words, we evaluated the AUC, with and without pre-training, for each group separately, for each model and pre-training target. The outcome incidence is summarized for each group and split in Table 6. The results of the stratification are illustrated in Figs 6 and 7. We observe a general improvement from transfer learning in all cases except the RN-33M model pre-trained on sex, for patients over 74 years, which had a slight performance reduction when compared to training from scratch. It seems that the improvement from transfer learning is generally larger for women than for men, and for younger people compared to older people. Interestingly, the best performing model overall (RN-900k) appears to perform

**Table 6**. AMI incidence.

| Subgroup | Train | Validation | Test |
|---|---|---|---|
| All, % | 5.8 | 5.7 | 5.8 |
| Men, % | 7.3 | 7.3 | 7.2 |
| Women, % | 4.1 | 3.9 | 4.3 |
| Under 46 years, % | 0.6 | 0.8 | 0.8 |
| 46–60 years, % | 4.5 | 4.1 | 4.4 |
| 61–73 years, % | 7.9 | 6.9 | 8.0 |
| Over 74 years, % | 9.7 | 10.5 | 9.8 |

Incidence of AMI for subgroups of the target dataset, for the different data splits. The age subgroups are quartiles.

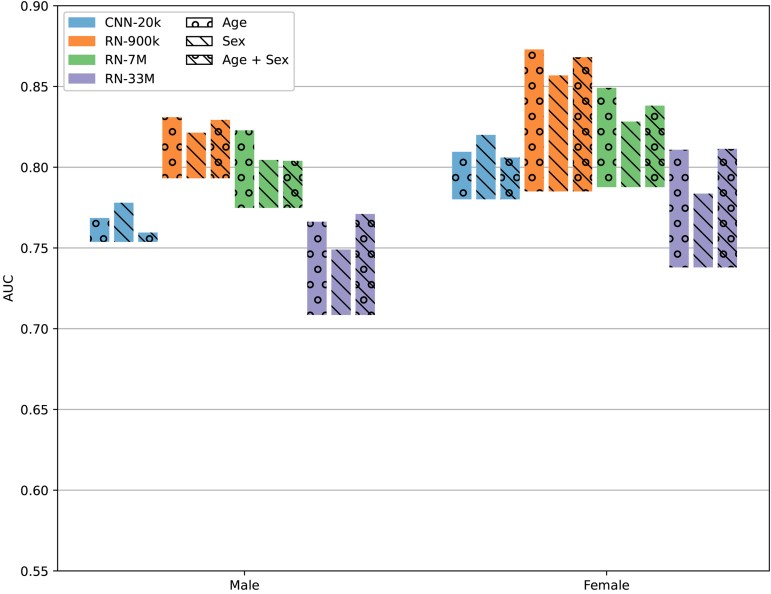

**Fig 6. Bar chart showing the AUC for predicting AMI, micro-averaged over both test sets, stratified on sex.** The bottom of each bar shows the AUC for models without pre-training, and the top of each bar shows the AUC for pre-trained models, using the full source dataset. The size of each bar thus corresponds to the improvement from transfer learning. The colors indicate the model, and the circles and stripes indicate the pre-training task.

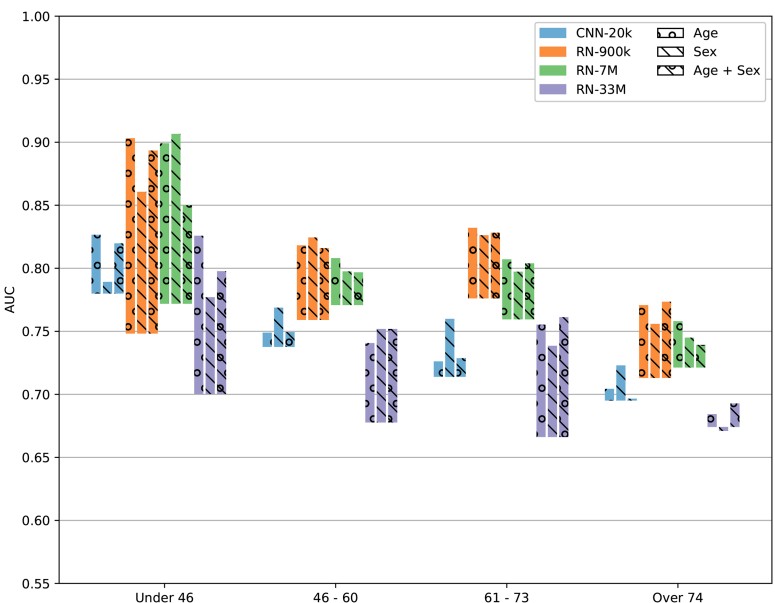

**Fig 7. Bar chart showing the AUC for predicting AMI, micro-averaged over both test sets, stratified on age quartiles.** The bottom of each bar shows the AUC for models without pre-training, and the top of each bar shows the AUC for pre-trained models, using the full source dataset. The size of each bar thus corresponds to the improvement from transfer learning. The colors indicate the model, and the circles and stripes indicate the pre-training task.

somewhat worse on women than men when not using transfer learning, and likewise slightly better on the older patients, but once we use transfer learning, the trend is reversed. Another observation is that the groups with larger benefits from transfer learning are those with lower incidence (i.e. women and young people). A possible explanation could be that the source material has much more data on "normal" ECGs than it does on pathological ECGs, and is thus more useful when the incidence is low.

Overall, our findings indicate that transfer learning might be useful for improving model fairness in groups with lower incidence. This may represent a fruitful avenue for further research into equitable deployment of clinical decision support tools based on deep learning.

## Shortcut learning results

We trained a random forest model to predict AMI using potential shortcut features from ECG-machine related metadata, including device location, software version, and cart ID. The model performed no better than random guessing (AUC 0.495) on the test set, dispelling our worries about ECG-machine related shortcuts. The metadata is described in more detail in Sect B in S1 Appendix.

Additionally, we trained a logistic regression model to predict AMI using age and sex as features, together with the predictions from the different models. The results are summarized in Table 7, which shows that pre-training leads to better predictions even when age and sex are directly included as features. This suggests that the benefit from transfer learning was not simply due to the ECG models inferring the age and sex of the patients, since that would be redundant in the presence of those features. In other words, age and sex do not appear to be shortcuts.

**Table 7. Results of logistic regression models assessing age and sex as potential shortcuts.**

| | Pre-training target | | | |
|---|---|---|---|---|
| | None | Age | Sex | Age & Sex |
| CNN-20k | 0.784 | 0.795 | 0.809 | 0.792 |
| RN-900k | 0.814 | 0.850 | 0.843 | 0.853 |
| RN-7M | 0.806 | 0.841 | 0.828 | 0.826 |
| RN-33M | 0.758 | 0.800 | 0.783 | 0.803 |

Results (AUC) of Logistic Regression models trained to predict AMI using age, sex, and ECG-model predictions as features. The rows and columns correspond to the ECG model architecture and pre-training target, respectively.

## Usability in the context of current care

It is important to note that the prediction models described in this paper are intended primarily as a basis for clinical decision support tools, and that clinicians must incorporate additional information beyond what is available to the model. To this end, it is important that clinicians are well-informed about the capabilities and limitations of the underlying model. Ideally, the decision support tool should be integrated into the clinical workflow in such a way that manual data inputs are avoided. Prospective clinical trials are required to assess patient outcomes in practice.

## Limitations

In this study, we have not attempted to adjust the hyperparameters of the models, which can be viewed as a limitation. Instead, we opted to use a selection of recently published model architectures, complete with all the specified settings, using them as is, even though they were tuned to different datasets on different tasks. This approach is not necessarily optimal if one wants to maximize the performance of a specific model for a given task. In our case, however, the idea was never to maximize the performance per se, but rather to assess the benefit of transfer learning across a range of models on our specific downstream task. We expect that all the models we used in this project would benefit to varying degrees by further optimizing the hyperparameters, but in order to get a truly fair comparison between the models, we would have to tune the parameters for each task, model and dataset size separately, which would be prohibitively expensive. The magnitude and consistency of our results across models suggests that the general trend in terms of benefit from transfer learning would remain even if all the relevant hyperparameters were optimized for each model.

A proper analysis of the relationship between model size and transfer learning is hindered in this study by fundamental differences in model architecture, primarily between the RN-900k model and the RN-7M and RN-33M models. Although the RN-900k model is much smaller in terms of number of parameters, it outperforms the other models in most situations. However, some of that performance is likely due to clever tricks being employed by the RN-900k model, perhaps most importantly the data augmentation and ensembling, which acts as an effective form of regularization. Incorporating these ideas into the other models would require substantial adjustments contrary to our initial philosophy of using these state-of-the-art models as is, but it would nevertheless be a promising future research direction. We note in passing that the evaluated architectures and ideas may profit from cross-fertilization – e.g., the ensemble and data augmentation techniques used in the RN-900k network could potentially be employed in the RN-7M and RN-33M architectures. Similarly, the Squeeze-and-Excitation blocks used in the RN-33M architecture could potentially be employed in the

RN-900k network. Future research would ideally be able to utilize more harmonized model baselines in order to better elucidate the effects of transfer learning.

When prediction models are used to estimate individual patient risks (as opposed to ranking patients by risk), issues of calibration may be important. However, in this paper we have opted not to investigate calibration, since it is orthogonal to our primary research question of discriminative ability. Depending on the specifics of the clinical applications, further research would be warranted to better understand the effects of transfer learning on model calibration.

Transfer learning is a broad topic that encompasses many different methods and strategies. In this paper, we have limited our focus to a single and relatively simple strategy. It is entirely possible that other approaches to transfer learning, including the self-supervised algorithms favored by Mehari et al. [22], would have performed even better. A comparison between different methods would be a welcome topic for future research.

## Summary and conclusions

In this study, we compared three different recently published state-of-the-art ResNet models and a small baseline convolutional neural network on the task of predicting AMI using ECGs. We explored the effects of a simple supervised transfer learning approach in which models were first pre-trained to predict age and/or sex on a separate collection of unrelated ECGs (lacking the AMI outcome label), and then fine-tuned to the target task of predicting AMI. This transfer learning scheme consistently improved our downstream predictions for all models, effectively increasing the maximum viable model size.

The approach also generalized to an independent ECG database (PTBXL), demonstrating robustness both with respect to population shift as well as nuances of outcome definitions, further strengthening the utility of the strategy.

Although all models were improved by pre-training, the smallest model improved the least, and the largest model, while benefiting the most from transfer learning, was still outperformed in absolute numbers by a substantially smaller model, underlining the important lesson that larger models are not always better.

Our results show that transfer learning can have a substantial positive impact on predicting AMI using ECGs. The simplicity and generalizability of using age and sex for pre-training, coupled with the fact that these variables are typically recorded in conjunction with the ECG, makes our approach an attractive and easy to implement first step when considering transfer learning for predicting AMI. Further research is required to appropriately compare our approach to more complicated methods, such as the self-supervised methods proposed in [22].

## Supporting information

**S1 Appendix. This file contains the results of additional evaluation metrics, a description of the ECG-machine metadata features, detailed model parameters, model architectures, and training times.**
(PDF)

## Author contributions

**Conceptualization:** Axel Nyström, Anders Björkelund, Mattias Ohlsson.

**Data curation:** Axel Nyström, Anders Björkelund.

**Formal analysis:** Axel Nyström.

**Funding acquisition:** Mattias Ohlsson, Jonas Björk, Ulf Ekelund.

**Investigation:** Axel Nyström.

**Methodology:** Axel Nyström, Mattias Ohlsson.

**Project administration:** Mattias Ohlsson, Jonas Björk, Ulf Ekelund, Jakob Lundager Forberg.

**Resources:** Ulf Ekelund.

**Software:** Axel Nyström.

**Supervision:** Anders Björkelund, Mattias Ohlsson, Jakob Lundager Forberg.

**Validation:** Axel Nyström, Anders Björkelund, Mattias Ohlsson.

**Visualization:** Axel Nyström, Anders Björkelund.

**Writing – original draft:** Axel Nyström.

**Writing – review & editing:** Axel Nyström, Anders Björkelund, Mattias Ohlsson, Jonas Björk, Ulf Ekelund, Jakob Lundager Forberg.

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
