## [Decision Letter · Decision Letter 0]

13 Mar 2025

PDIG-D-24-00450Transfer learning for predicting acute myocardial infarction using electrocardiogramsPLOS Digital Health Dear Dr. Nyström, Thank you for submitting your manuscript to PLOS Digital Health. After careful consideration, we feel that it has merit but does not fully meet PLOS Digital Health's publication criteria as it currently stands. Therefore, we invite you to submit a revised version of the manuscript that addresses the points raised during the review process. Please submit your revised manuscript within 60 days May 12 2025 11:59PM. If you will need more time than this to complete your revisions, please reply to this message or contact the journal office at digitalhealth@plos.org. Please include the following items when submitting your revised manuscript:* A rebuttal letter that responds to each point raised by the editor and reviewer(s). You should upload this letter as a separate file labeled 'Response to Reviewers'. This file does not need to include responses to any formatting updates and technical items listed in the 'Journal Requirements' section below.* A marked-up copy of your manuscript that highlights changes made to the original version. You should upload this as a separate file labeled 'Revised Manuscript with Track Changes'.* An unmarked version of your revised paper without tracked changes. You should upload this as a separate file labeled 'Manuscript'. If you would like to make changes to your financial disclosure, competing interests statement, or data availability statement, please make these updates within the submission form at the time of resubmission. Guidelines for resubmitting your figure files are available below the reviewer comments at the end of this letter. We look forward to receiving your revised manuscript. Kind regards, Ludwig Christian Giuseppe Hinske, M.D.Academic EditorPLOS Digital Health Ludwig Christian HinskeAcademic EditorPLOS Digital Health Leo Anthony CeliEditor-in-ChiefPLOS Digital Healthorcid.org/0000-0001-6712-6626 **Journal Requirements:**

1. We ask that a manuscript source file is provided at Revision. Please upload your manuscript file as a .doc, .docx, .rtf or .tex.

 **Additional Editor Comments (if provided):****Reviewers' Comments:** Reviewer's Responses to Questions

**Comments to the Author**

1. Does this manuscript meet PLOS Digital Health’s publication criteria? Is the manuscript technically sound, and do the data support the conclusions? The manuscript must describe methodologically and ethically rigorous research with conclusions that are appropriately drawn based on the data presented.

Reviewer #1: Yes

Reviewer #2: Yes

Reviewer #3: Yes

2. Has the statistical analysis been performed appropriately and rigorously?

Reviewer #1: Yes

Reviewer #2: N/A

Reviewer #3: Yes

3. Have the authors made all data underlying the findings in their manuscript fully available (please refer to the Data Availability Statement at the start of the manuscript PDF file)?

Reviewer #1: No

Reviewer #2: No

Reviewer #3: Yes

4. Is the manuscript presented in an intelligible fashion and written in standard English?

Reviewer #1: Yes

Reviewer #2: Yes

Reviewer #3: Yes

5. Review Comments to the Author

Reviewer #1: Nyström et al. present a very well designed and described large study to predict acute myocardial infarction in the emergency department using electrocardiograms and transfer learning. The findings are relevant from a methodological perspective, but could be strengthened from a clinical perspective.

1) I suggest the authors include more information on the ‘Usability of the model in the context of current care’ (item 27 in the TRIPOD+AI statement). Even if the implementation is beyond the scope of this paper, the ultimate goal of a clinical prediction model should be to do so, so discussion of attention points would be relevant in my opinion.

2) From a clinical perspective, when prediction models are used on an individual basis often with some threshold values triggering actions (e.g. if risk is over 10%, then …), calibration of the model describing agreement between predicted risk and observed events is crucial. (This aspect is way underreported in the machine learning literature.) I suggest the authors report calibration plots and metrics like calibration-in-the-large and calibration slope (see e.g. CalibrationCurves R package, not sure about corresponding tools in Python).

3) It would be clinically relevant to investigate the results from an algorithmic fairness perspective (TRIPOD+AI items 3c, 14, 23a). Given the well-known sex difference in cardiovascular risk, also mentioned by the authors, that would be an obvious sensitive attribute or subgrouping to report the results by and investigate the fairness of the model. Would there be other relevant factors (e.g. socio-economic status, ethnicity, if there is data on these)? Useful resources/references on the topic (not specific to ECG):

Tibor V Varga - Algorithmic fairness in cardiovascular disease risk prediction: overcoming inequalities: Open Heart 2023;10:e002395.

Chen, R.J., Wang, J.J., Williamson, D.F.K. et al. Algorithmic fairness in artificial intelligence for medicine and healthcare. Nat. Biomed. Eng 7, 719–742 (2023). https://doi.org/10.1038/s41551-023-01056-8

Ricci Lara, M.A., Echeveste, R. & Ferrante, E. Addressing fairness in artificial intelligence for medical imaging. Nat Commun 13, 4581 (2022). https://doi.org/10.1038/s41467-022-32186-3

4) Reporting model performance by sex (just the main results, not for different %s of source and target data) could also shed light on another aspect that has raised concerns around deep learning models recently i.e. shortcut learning. Given the good discrimination for sex, and higher cardiovascular risk among male patients, I wonder how much are the results explained by the model predicting sex as a shortcut and not the actual task i.e. AMI. It would be relevant to see the distribution of the outcome not only by splits, but also by sex within the splits and if the authors think that there could be any other similar factors. These tables could be part of the supplementary materials. Please consider what other shortcuts would be feasible in the dataset and discuss their potential impact on the results. E.g. Are different ECG devices used in different hospitals where the risk of AMI is lower/higher? Can the type of the machine be detected? Useful resources/references on the topic:

Ong Ly, C., Unnikrishnan, B., Tadic, T. et al. Shortcut learning in medical AI hinders generalization: method for estimating AI model generalization without external data. npj Digit. Med. 7, 124 (2024). https://doi.org/10.1038/s41746-024-01118-4

Geirhos, R., Jacobsen, JH., Michaelis, C. et al. Shortcut learning in deep neural networks. Nat Mach Intell 2, 665–673 (2020). https://doi.org/10.1038/s42256-020-00257-z

5) From an epidemiological perspective, I am curious what happened to the people in the analysis who died within 30 days? How many were there and how were they handled in the analysis if the cause of death was / was not AMI?

6) Also, were people/visits with missing data different from those without? What can be the main reasons for missingness and could this affect the generalizability of the results?

Reviewer #2: This paper proposes a straightforward application of transfer learning. The pre-trained models are initially trained to classify sex and age using a collection of 840k ECGs from non-chest-pain patients and are then fine-tuned to predict AMI using 44k ECGs from chest-pain patients.

Four different baseline models are studied and selected from four references [20, 25-27]. The model architecture details are reported in the supporting documents. An ablation study is also conducted to evaluate the effect of the dataset size on the tuned model's performance.

Transfer learning has a broad scope. However, only the fine-tuning strategy is studied and described in Lines 221-231. Other transfer learning strategies, such as knowledge distillation, should be discussed or even evaluated in experiments.

Additional performance metrics (e.g., precision, recall) should be reported in experiments besides the currently reported AUC and accuracy.

As acknowledged in the current paper (Line 319), the choice of hyperparameters is not studied. Is there any special reason for this?

Reviewer #3: The paper utilizes transfer learning to predict acute myocardial infarction (AMI) from ECGs. The study benefits from access to 840k ECGs for pretraining and 44k for AMI fine-tuning. However, it does not employ state-of-the-art ECG foundation models that have been pre-trained on large-scale ECG datasets using self-supervised learning (SSL) methods or domain-specific pretraining techniques. Instead, the authors perform a relatively simple supervised pretraining on sex and age classification before fine-tuning for AMI detection.

Major concerns:

1. Outdated Transfer Learning Approach

- Relies solely on supervised pretraining for sex and age classification instead of modern self-supervised learning (SSL) or contrastive learning approaches.

- No comparison with ECG foundation models such as:

- PTB-XL pre-trained models

- Self-supervised ECG models (Mehari et al. 2022, 2023)

- Contrastive Predictive Coding (CPC) for ECGs

- ECG-BERT models

- Time-series Transformers for ECG (Hyena, Perceiver-IO, etc.)

2. Weak Pretraining Task Choice (Age and Sex)

- Age and sex classification does not provide meaningful ECG features for AMI prediction.

- Self-supervised learning captures physiological representations better than simple demographic features.

3. Lack of Clinical Interpretability and Explainability

- No analysis on feature importance or interpretability (e.g., DeepSHAP, Grad-CAM, attention maps).

- Could benefit from:

- Saliency map visualizations

- SHAP value analysis

- Comparison with physician interpretations

4. Lack of Robustness Tests

- No evaluation of generalization across demographics, hospitals, or ECG devices.

- Does not assess performance under missing leads or noisy ECGs.

5. Lack of External Validation

- Entirely based on Skåne Emergency Medicine (SEM) database.

- No validation on independent datasets (e.g., PTB-XL, MIMIC-III, CODE-15).

6. Poor Comparison to Non-Transfer Learning Baselines

- Only compares to models trained without pretraining.

- Should compare to:

- End-to-end models trained directly on AMI data

- Models pre-trained using self-supervised or contrastive learning methods

6. PLOS authors have the option to publish the peer review history of their article (what does this mean?). If published, this will include your full peer review and any attached files.

**Do you want your identity to be public for this peer review?** For information about this choice, including consent withdrawal, please see our Privacy Policy.

Reviewer #1: **Yes: **Adam Hulman

Reviewer #2: No

Reviewer #3: No

---

## [Decision Letter · Decision Letter 1]

25 Aug 2025

PDIG-D-24-00450R1Transfer learning for predicting acute myocardial infarction using electrocardiogramsPLOS Digital Health Dear Dr. Nyström, Thank you for submitting your manuscript to PLOS Digital Health. After careful consideration, we feel that it has merit but does not fully meet PLOS Digital Health's publication criteria as it currently stands. Therefore, we invite you to submit a revised version of the manuscript that addresses the points raised during the review process. Please submit your revised manuscript within 30 days Sep 24 2025 11:59PM. If you will need more time than this to complete your revisions, please reply to this message or contact the journal office at digitalhealth@plos.org. Please include the following items when submitting your revised manuscript:* A rebuttal letter that responds to each point raised by the editor and reviewer(s). You should upload this letter as a separate file labeled 'Response to Reviewers'. This file does not need to include responses to any formatting updates and technical items listed in the 'Journal Requirements' section below.* A marked-up copy of your manuscript that highlights changes made to the original version. You should upload this as a separate file labeled 'Revised Manuscript with Track Changes'.* An unmarked version of your revised paper without tracked changes. You should upload this as a separate file labeled 'Manuscript'. If you would like to make changes to your financial disclosure, competing interests statement, or data availability statement, please make these updates within the submission form at the time of resubmission. Guidelines for resubmitting your figure files are available below the reviewer comments at the end of this letter. We look forward to receiving your revised manuscript. Kind regards, Ludwig Christian Giuseppe Hinske, M.D.Academic EditorPLOS Digital Health Ludwig Christian HinskeAcademic EditorPLOS Digital Health Leo Anthony CeliEditor-in-ChiefPLOS Digital Healthorcid.org/0000-0001-6712-6626  **Journal Requirements:** If the reviewer comments include a recommendation to cite specific previously published works, please review and evaluate these publications to determine whether they are relevant and should be cited. There is no requirement to cite these works unless the editor has indicated otherwise.  **Additional Editor Comments (if provided):****Reviewers' Comments:** Reviewer's Responses to Questions

**Comments to the Author**

1. If the authors have adequately addressed your comments raised in a previous round of review and you feel that this manuscript is now acceptable for publication, you may indicate that here to bypass the “Comments to the Author” section, enter your conflict of interest statement in the “Confidential to Editor” section, and submit your "Accept" recommendation.

Reviewer #1: (No Response)

Reviewer #2: All comments have been addressed

Reviewer #3: (No Response)

2. Does this manuscript meet PLOS Digital Health’s publication criteria? Is the manuscript technically sound, and do the data support the conclusions? The manuscript must describe methodologically and ethically rigorous research with conclusions that are appropriately drawn based on the data presented.

Reviewer #1: Yes

Reviewer #2: Yes

Reviewer #3: (No Response)

3. Has the statistical analysis been performed appropriately and rigorously?

Reviewer #1: Yes

Reviewer #2: N/A

Reviewer #3: (No Response)

4. Have the authors made all data underlying the findings in their manuscript fully available (please refer to the Data Availability Statement at the start of the manuscript PDF file)?

Reviewer #1: No

Reviewer #2: No

Reviewer #3: (No Response)

5. Is the manuscript presented in an intelligible fashion and written in standard English?

Reviewer #1: Yes

Reviewer #2: Yes

Reviewer #3: (No Response)

6. Review Comments to the Author

Reviewer #1: The authors have addressed my comments in the revised manuscript. I find the added results highly interesting and the manuscript more clear. I have one minor comment left:

Although the presented analysis and results are interesting (and reassuring for metadata), I am not sure why the authors took a different approach for age and sex. To be honest, I am not 100% sure what the latter approach is, so this could be explained more both in the Methods and the Table 7 legend (as the table should be self-explanatory). Are age and sex combined with the predicted probabilities as features? What I had in mind was much simpler and similar to what they did with the metadata i.e. to fit a logistic regression with age and sex as the sole predictors to create a benchmark performance and see whether deep learning, with or without transfer learning, beats that as expected.

Reviewer #2: The revision is fine; the authors have acknowledged the limitations of the study, such as the lack of evaluation of modern transfer learning strategies.

Reviewer #3: This paper investigates the utility of supervised transfer learning to improve deep learning model performance in predicting AMI using ECGs in emergency department patients. The authors pre-train models to predict age and/or sex using a large dataset of 840K ECGs from non-chest-pain patients and then fine-tune them for AMI prediction using 44K ECGs from chest-pain patients. They compare multiple CNN architectures, including 3 ResNet variants, and show consistent improvements in AUC across all models with transfer learning, achieving up to 0.85 AUC compared to 0.79 without pre-training. The study further explores how source and target dataset sizes affect performance, examines generalizability on the PTB-XL dataset, evaluates shortcut learning risks, and conducts subgroup and fairness analyses. The results demonstrate that simple supervised pre-training on age and sex substantially enhances AMI prediction performance and generalizes well across datasets.

Major Comments

1. It is crucial to confirm that there is no patient overlap between the pre-training (source) and fine-tuning (target) sets. While the authors mention that patients in the source dataset were excluded from the target, stronger emphasis on this separation is warranted, especially due to the known confounding role of age and sex in AMI risk. This is necessary to eliminate potential data leakage and ensure fair generalization.

2. The choice of AUC as the primary evaluation metric for AMI prediction, despite the class imbalance (only ~6% AMI prevalence), is suboptimal. Precision-recall AUC would better reflect performance in this imbalanced setting. Consider adding precision-recall curves or PR-AUC as a complementary metric.

3. The manuscript would benefit from a clearer comparative discussion of how the proposed supervised pre-training (age/sex prediction) fares against unsupervised (e.g., autoencoder by Jang et al [16], self-supervised or contrastive pre-training methods (e.g., [21]). Could the authors quantify or qualitatively discuss relative benefits or limitations of supervised vs. un/self-supervised strategies in this context?

4. The performance comparison across different models may be confounded by differing levels of architectural optimization and pre-training compatibility. This difference should be highlighted as a limitation and could be addressed in future work via more harmonized model baselines.

5. The external validation on PTB-XL is appreciated, but the results could be further elaborated. For example, how consistent is the model performance across label distributions in PTB-XL vs SEM?

6. Subgroup analysis reveals interesting trends. These insights should be explored further, especially considering the low AMI incidence in these groups. What might be the implications for clinical deployment in underrepresented populations?

7. The shortcut learning analysis is an interesting component. However, I do not see the conclusion of “the predictions from the pre-trained models improved the logistic regression model” is made. In Table 7, you have “No pre-training”, “Age”, “Sex”, and “Age & “Sex”. Are the latter 3 the resulting accuracy of logistic regression? Since you have the 4 deep learning models as 4 rows, don’t you need to have logistic regression as the fifth row for comparison?

Minor Comments

1. Minor grammar/wording issues in places (e.g., 'the patients in the source dataset is...' should be 'are').

2. In the exclusion diagram, it would be helpful to clarify the logic in plain language.

3. Consider including model training time or FLOPs in Table 2 to contextualize complexity-performance trade-off.

4. Several useful details are in the S1 Appendix. Reference them more clearly in the main text.

5. What's the self-supervised learning objective in Mehari et al [22] S4 pre-training regime?

7. PLOS authors have the option to publish the peer review history of their article (what does this mean?). If published, this will include your full peer review and any attached files.

**Do you want your identity to be public for this peer review?** For information about this choice, including consent withdrawal, please see our Privacy Policy.

Reviewer #1: **Yes: **Adam Hulman

Reviewer #2: No

Reviewer #3: None

 **Figure resubmission:**  While revising your submission, we strongly recommend that you use PLOS’s NAAS tool (https://ngplosjournals.pagemajik.ai/artanalysis) to test your figure files. NAAS can convert your figure files to the TIFF file type and meet basic requirements (such as print size, resolution), or provide you with a report on issues that do not meet our requirements and that NAAS cannot fix. 

After uploading your figures to PLOS’s NAAS tool - https://ngplosjournals.pagemajik.ai/artanalysis, NAAS will process the files provided and display the results in the "Uploaded Files" section of the page as the processing is complete. If the uploaded figures meet our requirements (or NAAS is able to fix the files to meet our requirements), the figure will be marked as "fixed" above. If NAAS is unable to fix the files, a red "failed" label will appear above. When NAAS has confirmed that the figure files meet our requirements, please download the file via the download option, and include these NAAS processed figure files when submitting your revised manuscript. **Reproducibility:** To enhance the reproducibility of your results, we recommend that authors of applicable studies deposit laboratory protocols in protocols.io, where a protocol can be assigned its own identifier (DOI) such that it can be cited independently in the future. Additionally, PLOS ONE offers an option to publish peer-reviewed clinical study protocols. Read more information on sharing protocols at https://plos.org/protocols?utm_medium=editorial-email&utm_source=authorletters&utm_campaign=protocols

---

## [Editor Report · Decision Letter 2]

6 Oct 2025

Transfer learning for predicting acute myocardial infarction using electrocardiograms

PDIG-D-24-00450R2

Dear Dr. Nyström,

We're pleased to inform you that your manuscript has been judged scientifically suitable for publication and will be formally accepted for publication once it meets all outstanding technical requirements.

Within one week, you'll receive an e-mail detailing the required amendments. When these have been addressed, you'll receive a formal acceptance letter and your manuscript will be scheduled for publication.

An invoice for payment will follow shortly after the formal acceptance. To ensure an efficient process, please log into Editorial Manager at https://www.editorialmanager.com/pdig/ click the 'Update My Information' link at the top of the page, and double check that your user information is up-to-date. For billing related questions, please contact billing support at https://plos.my.site.com/s/.

Kind regards,

Ludwig Christian Giuseppe Hinske, M.D.

Academic Editor

PLOS Digital Health